# Demonstrating paths for unlocking the value of cloud genomics through cross cohort analysis

Nicole Deflaux [1,16], Margaret Sunitha Selvaraj [2,3,4,5,16], Henry Robert Condon[6], Kelsey Mayo [7], Sara Haidermota[2,8], Melissa A. Basford[7], Chris Lunt [9], Anthony A. Philippakis [10], Dan M. Roden[11,12,13], Joshua C. Denny[9], Anjene Musick[9], Rory Collins[14,15], Naomi Allen [14,15], Mark Effingham [15], David Glazer [1], Pradeep Natarajan [2,3,4,5,8] & Alexander G. Bick [6] ✉

Recently, large scale genomic projects such as *All of Us* and the UK Biobank have introduced a new research paradigm where data are stored centrally in cloud-based Trusted Research Environments (TREs). To characterize the advantages and drawbacks of different TRE attributes in facilitating cross-cohort analysis, we conduct a Genome-Wide Association Study of standard lipid measures using two approaches: meta-analysis and pooled analysis. Comparison of full summary data from both approaches with an external study shows strong correlation of known loci with lipid levels ($R^2 \sim 83–97\%$). Importantly, 90 variants meet the significance threshold only in the meta-analysis and 64 variants are significant only in pooled analysis, with approximately 20% of variants in each of those groups being most prevalent in non-European, non-Asian ancestry individuals. These findings have important implications, as technical and policy choices lead to cross-cohort analyses generating similar, but not identical results, particularly for non-European ancestral populations.

Traditional data sharing processes require researchers to download copies of data to their own systems. More recently, health research is shifting to use Trusted Research Environments (TREs), such as the *All of Us* Researcher Workbench (AoU RW) and the UK Biobank Research Analysis Platform (UKB RAP), for large-scale clinical and genomic data-sharing and analysis[1–4]. In general, a TRE is a secure computing environment which provides approved researchers with tools to access and analyze sensitive health data. TREs offer many benefits, including (1) increased protection of study participant data, (2) decreased barriers to access and analyze data, (3) lower cost of shared data storage, and (4) increased collaboration across the scientific community[5–7]. The positive impact of TREs is clear, as is their potential to facilitate population- and global-scale health research[8,9].

[1]Verily Life Sciences, San Francisco, CA, USA. [2]Program in Medical and Population Genetics and the Cardiovascular Disease Initiative, Broad Institute of Harvard and MIT, Cambridge, MA, USA. [3]Department of Medicine, Harvard Medical School, Boston, MA, USA. [4]Cardiovascular Research Center, Massachusetts General Hospital, Boston, MA, USA. [5]Center for Genomic Medicine, Massachusetts General Hospital, Boston, MA, USA. [6]Division of Genetic Medicine, Vanderbilt University Medical Center, Nashville, TN, USA. [7]Vanderbilt Institute for Clinical and Translational Research, Vanderbilt University Medical Center, Nashville, TN, USA. [8]Division of Cardiology, Massachusetts General Hospital, Boston, MA, USA. [9]All of Us Research Program, National Institutes of Health, Bethesda, MD, USA. [10]Broad Institute of Harvard and MIT, Cambridge, MA, USA. [11]Department of Medicine, Vanderbilt University Medical Center, Nashville, TN, USA. [12]Department of Pharmacology, Vanderbilt University Medical Center, Nashville, TN, USA. [13]Department of Biomedical Informatics, Vanderbilt University Medical Center, Nashville, TN, USA. [14]Nuffield Department of Population Health, University of Oxford, Oxford, Oxfordshire, UK. [15]UK Biobank, Cheadle, Stockport, UK. [16]These authors contributed equally: Nicole Deflaux, Margaret Sunitha Selvaraj. ✉e-mail: alexander.bick@vumc.org

For many important reasons, including participant data privacy, trust and security, TREs often implement a variety of policy and technological safeguards. For example, data that reside in an enclave may not be allowed to leave the environment in non-aggregated form[10,11]. Researchers wishing to safely and appropriately analyze data across different TREs face technological hurdles and policy requirements to do so[12]. Several approaches to data analysis across enclaves have been proposed. These include a meta-analysis whereby researchers perform analysis in separate TREs and then meta-analyze de-identified results outside of an enclave, and pooled analysis whereby researchers create and analyze merged data within a single enclave (Fig. 1). Each approach has advantages and limitations. All approaches to cross-analysis benefit from improved harmonization and standardization of data, policies, and working environments[8,13]. Together with the broader research community, data providers play a critical role in charting approved paths to cross-analysis and disseminating this information broadly. This paper describes approaches to cross-analyze *All of Us* and UK Biobank data, and discusses benefits

and limitations of each approach with respect to cost, complexity, and scientific utility (Supplementary Fig. 1).

Specifically, a genome-wide association study (GWAS) was used to explore cross-analysis of UK Biobank and *All of Us* data, as it is a standard analytical approach that benefits significantly from the boost in power obtained from increased sample size[14,15]. Additionally, methods for meta-analysis and pooled GWAS are well developed[16]. Circulating lipid concentrations were chosen as the target phenotype to enable validation of the two approaches by replicating well-established genetic associations. The work presented here is the result of collaboration between the *All of Us* and UK Biobank programs intended to build and describe research resources rather than discover novel associations.

## Results

We performed a genome-wide association study on circulating lipid levels involving *All of Us* whole genome sequence data and UK Biobank whole exome sequence data twice - (1) by meta-analyzing GWAS results

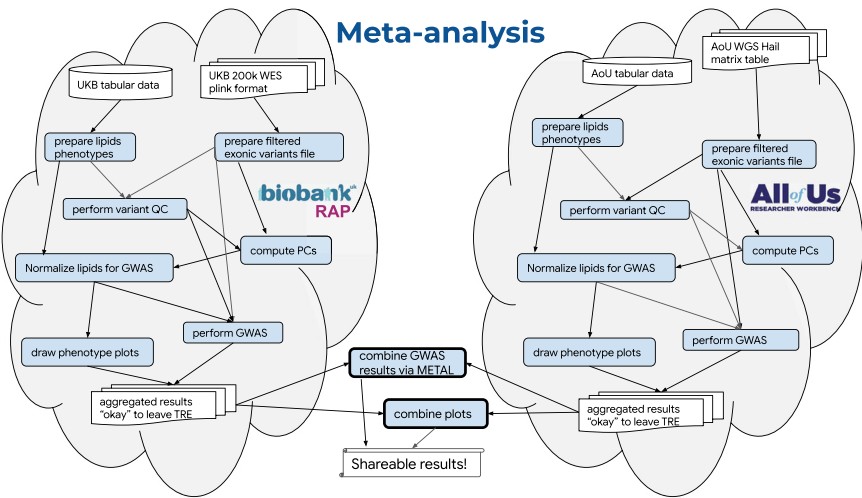

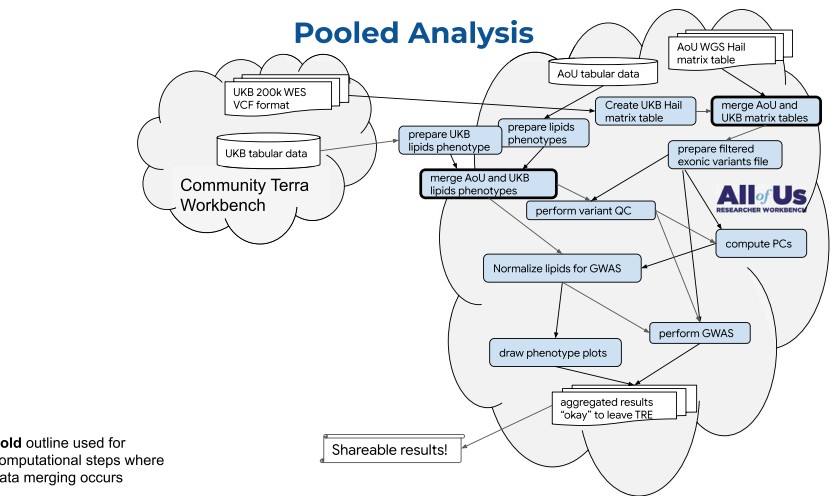

**Fig. 1 | Outline of steps in the meta- and pooled analyses for *All of Us* and UK Biobank cross-cohort analysis.** Researchers analyzing data across TREs, using either meta-analysis or a pooled approach, must negotiate policy requirements and technical hurdles. Bold outline is used for computational steps where data merging occurs. Top: Computational steps involved in meta-analysis, many of which are duplicated. Bottom: Computational steps involved in pooled analysis, where each distinct step is performed only once. *All of Us*, the *All of Us* logo, and "The Future of Health Begins with You" are service marks of the U.S. Department of Health and Human Services.

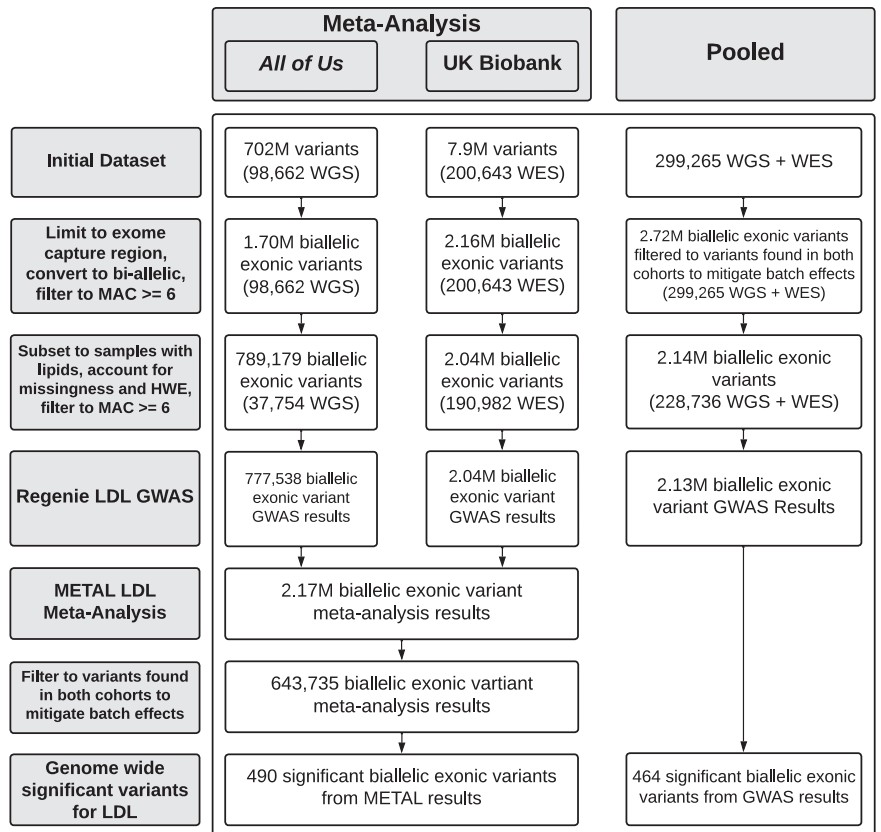

**Fig. 2 | Flow diagram highlighting the number of variants and sequenced samples retained at each stage of the meta- and pooled analyses.** Whole Genome Sequencing, WGS. Whole Exome Sequencing, WES. Minor Allele Count, MAC.

from separate TREs and (2) by analyzing pooled data in a single TRE. The goals, recruitment methods, scientific rationale and genomic data for *All of Us* and UK Biobank have been described previously[1,2]. In *All of Us*, we leveraged 98,622 whole genome sequenced samples alongside 200,643 whole exome sequenced samples from the UK Biobank. Although whole genome sequence data are available for UK Biobank, pooled analysis would require the data to be moved to a common enclave, which is not permitted by its access policy. The 200k exome release from UK Biobank was therefore explicitly chosen for use in this project because it was the last release of individual-level UK Biobank sequence data permitted to be analyzed outside of the UKB RAP, and therefore available for use in both pooled and meta-analyses performed on the AoU RW. Since our project was focused on comparing the computational approaches rather than on discovering new associations, maximal sample sizes were not needed.

## The meta-analysis

For the meta-analysis, GWAS of lipid levels were performed separately in the *All of Us* and UK Biobank TREs (see supplement for further details). Phenotypes were prepared separately. We curated lipid phenotypes (high-density lipoprotein cholesterol: HDL-C, low-density lipoprotein cholesterol: LDL-C, total cholesterol: TC, triglycerides: TG) using the cohort builder tool within the AoU RW. We obtained phenotype information on one or more lipid measurements from electronic health records for 37,754 *All of Us* participants with available whole genome sequence data. In the UK Biobank, one or more lipid measurements from systematic central laboratory assay were available for 190,982 participants with exome sequence data[17]. Covariate information (age, sex at birth, self-reported race) and data on lipid-lowering medication for these corresponding samples were extracted from *All of Us* survey and electronic health record data and UK Biobank self-reported data.

The lipid phenotypes were adjusted for statin medication[18,19] and normalized (see supplement).

A GWAS was performed in each cohort separately using REGENIE[20] on the subset of variants within the UK Biobank exonic capture regions (Fig. 2). In each TRE, we retained variants with allele count (AC) >=6, since variants with an exceptionally low allele count are not considered by the analysis method, and obtained 1,699,534 biallelic exonic variants from *All of Us* and 2,158,225 from the UK Biobank. After applying variant quality control to filter out low quality variants from the subset of samples in the lipids cohort, single variant GWAS was performed with 789,179 variants from the *All of Us* cohort and associated with the LDL-C phenotype. Separately, this same process was carried out with 2,037,169 variants from the UK Biobank cohort. Each set of results was then downloaded, keeping in mind that before dissemination they must be filtered to remove AC < 40 in accordance with the *All of Us* Data and Statistics Dissemination Policy, which disallows disclosure of group counts under 20 since a given individual could have two copies of a single allele[10]. *All of Us* does permit researchers to request an exception to this policy through the program's Resource Access Board, which we were granted for the results in this particular study. Finally, we meta-analyzed variants by combining the summary statistics obtained from both studies using an inverse variance-weighted fixed effects method implemented in METAL[21]. 490 variants from 321 loci ($r^2$:0.5) were significantly associated ($p < 5E-08$) with LDL-C (Fig. 3b, Supplementary Data 1).

## The pooled analysis

For the pooled analysis, data from the UK Biobank were copied into the AoU RW for cross-analysis with data from *All of Us*. Phenotypes were prepared as previously described and merged into a single table. Genomic data were prepared by merging variants for all available samples from the UK Biobank and *All of Us* cohorts into a

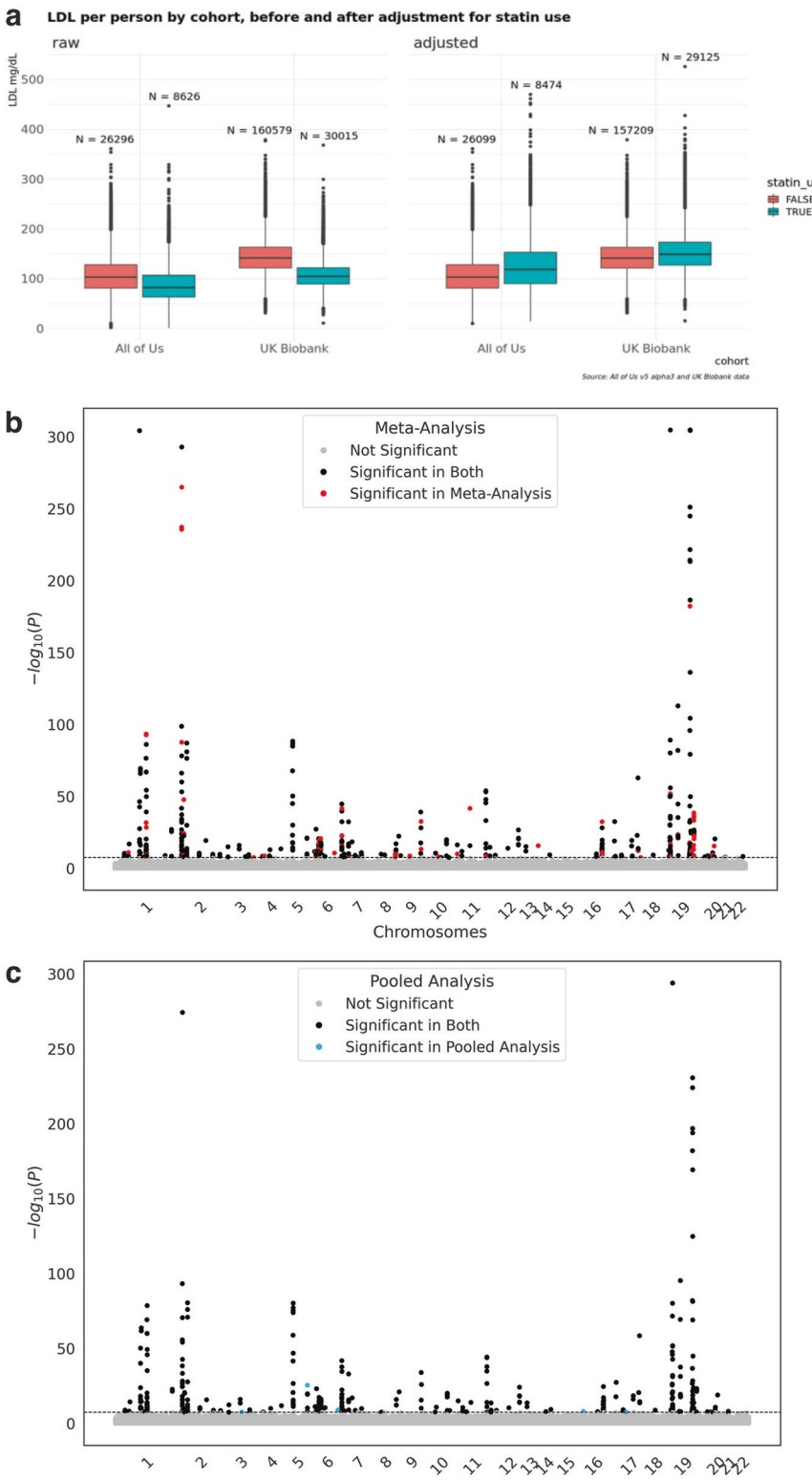

**Fig. 3 | GWAS phenotype and results. a** Participant LDL-C levels for each cohort, before (left) and after (right) adjusting for statin use. The black center line denotes the median value (50th percentile), while the boxes contain the 25th to 75th percentiles of data. The black whiskers mark the 5th and 95th percentiles, and values beyond these upper and lower bounds are considered outliers, marked with black dots. Note that a few very high outliers were filtered to improve readability of the plot. **b** Meta analysis results for LDL-C GWAS on merged exonic variants. **c** Pooled results for LDL-C GWAS on merged exonic variants. Both replicate known gene associations.

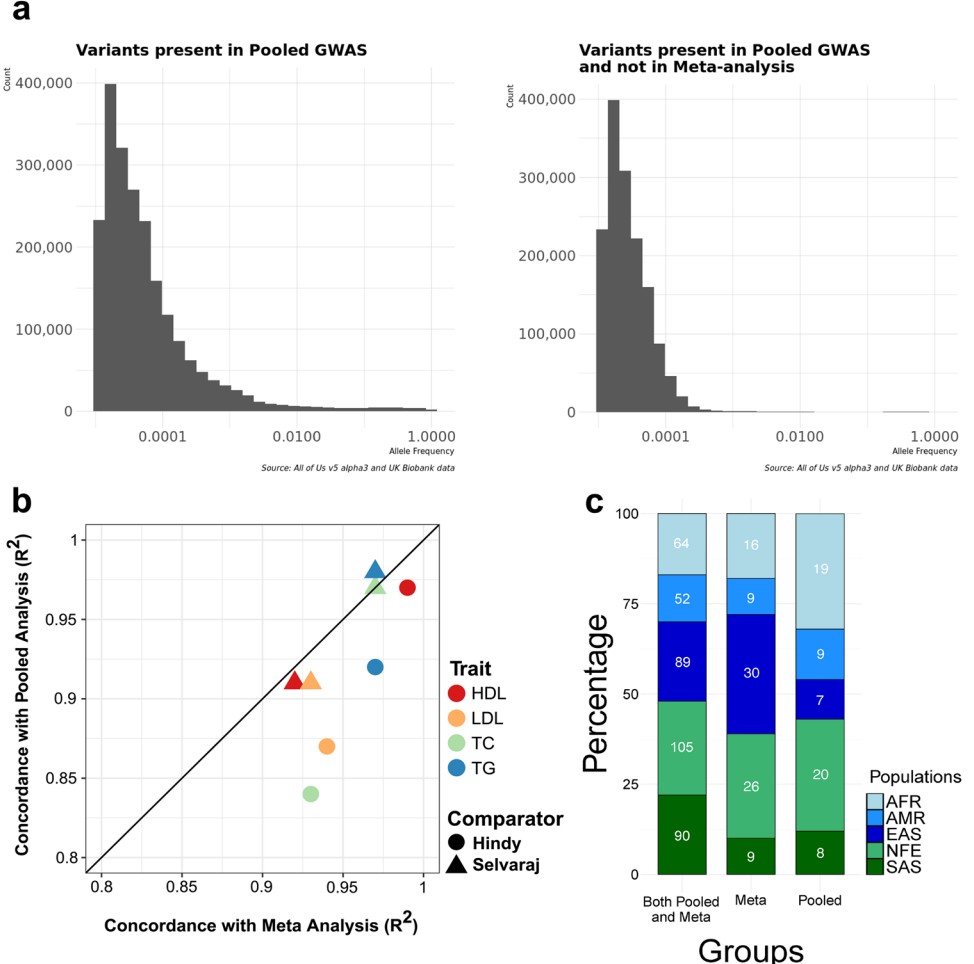

**Fig. 4 | Scientific differences in pooled and meta-analyses. a** Examination of variants included only in the pooled analysis. **b** Comparison of lipid GWAS results against two previously published reference datasets: Hindy[22] and Selvaraj[23]. HDL high-density lipoprotein cholesterol, LDL low-density lipoprotein cholesterol, TC total cholesterol, TG triglycerides (**c**) Bar chart of ancestry proportions across all methods with the variant results meeting genome-wide significance superimposed. Here, AFR, AMR, EAS, NFE, and SAS indicate African, American, East Asian, Non-Finish European, and South Asian ancestry groups, respectively.

single genomic data set (Fig. 2). For the pooled analysis, biallelic variants were retained if the same variant was present in both cohorts to avoid the clear batch effect of a variant present in only one cohort. We obtained 2,715,453 biallelic exonic variants for the pooled analysis after subsetting to UK Biobank exonic capture regions and filtering allele count (AC) >=6, since variants with an exceptionally low allele count are not considered by the analysis method. After applying variant quality control to filter out low quality variants from the subset of samples in the lipids cohort, single variant GWAS was performed with 2,135,845 merged variants in the pooled cohort for each of the lipid phenotypes. Cohort source (either *All of Us* or UK Biobank) was included as an additional covariate to mitigate potential batch effects from the different sequencing approaches and informatics pipelines used in *All of Us* and UK Biobank (see supplement). 464 variants were significantly associated ($p < 5E-08$) with the LDL-C phenotype from 284 loci ($r^2$:0.5) (Fig. 3c, Supplementary Data 2).

## Scientific differences between pooled and meta-analyses
We sought to test whether important scientific differences exist between our pooled and meta-analyses. We first investigated how the analytical approach impacted the identification of variants significantly associated with our phenotypes of interest. Most of the significant variants identified by either method were previously reported to be associated with plasma lipids in external datasets

(Supplementary Data 1 and 2). Of the novel significant variants, most were short insertions/deletions which were largely excluded from prior efforts. Gene prioritization of the GWAS results from our analysis, fine-mapped variants to genes important to lipids including *APOE*, *APOA2*, *LDLR*, *PCSK9*, *CEPT*, *APOA5*, *APOB* with top 20 prioritization scores. We then tested the extent to which each approach replicates known associations by comparing lipid GWAS results with two previously published datasets that contain the largest amount of data on exome and genome sequencing lipid associations[22,23]. The Selvaraj study includes diverse individuals from an external TOPMed cohort. The Hindy study included ~40,000 individuals from the UK Biobank (partially overlapping with our UK Biobank dataset) as well as ~170,000 other individuals, most of whom were of European ancestry. Effect sizes from both of our analyses are highly correlated with the two previously published standards (Fig. 4b). Analytical approach had little impact on either the number of significant SNPs or the concordance ($R^2$) of associations in common with the Selvaraj study. When compared with the Hindy study, an average of ~3 more genome-wide significant SNPs were retained with the pooled analysis (Supplementary Fig. 10), however the concordance ($R^2$) was slightly lower for all lipid phenotypes using the pooled approach (Fig. 4b). We next examined whether the pooled analysis includes a broader total set of variants than the meta-analysis. There are 1,496,404 variants which were present in only pooled analysis, most of which were of lower minor allele frequency (Fig. 4a).

**Table 1 | Rare variants uniquely significant in either meta-analysis or pooled analysis**

| Analysis Type | RS Id | AF | Ancestry | Gene-Mutation |
|---|---|---|---|---|
| Meta-analysis | rs72646508 | 0.002 | AFR | *PCSK9* p.Leu253Phe |
| Meta-analysis | rs145777339 | 0.003 | AMR | *APOB* p.Tyr3098= |
| Pooled | rs981175281 | 0.0002171081 | AFR | *PDZRN3* intron_variant |
| Pooled | rs150401820 | 0.0007244627 | AFR | *LRP4* p.Asp91Asp |
| Pooled | rs370601772 | 0.0004823927 | AFR | *MYO19* p.Lys118Asn |
| Pooled | rs121908030 | 0.0001933862 | AFR | *LDLR* p.Asp389Asn |
| Pooled | rs28942084 | 7.236588e-05 | AFR | *LDLR* p.Pro770Leu |
| Pooled | rs142412517 | 0.001 | AMR | APOE p.Arg239Trp |

Next, we tested how the analytical approach impacted the ancestry frequency distributions of significant variants. We obtained ancestry data from gnomAD and referenced the popmax ancestry information[24]. Out of the 490 significant variants from meta-analysis and 464 variants from pooled analysis, 400 variants were common between both analyses. The variants common between both analyses were from different ancestral groups, 16% African, 13% American, 26% Non-Finnish European, 22% each from East Asian and South Asian groups (Fig. 4c, Supplementary Data 3). Around 90 variants were identified as genome-wide significant in meta-analysis but not in the pooled analysis, whereas 64 variants were significant in the pooled analysis but not in meta-analysis. Some of the variants considered significant in only one method were below but near the significance cutoff, or not included in both analyses due to AC filtering or variant QC (Supplementary Figs. 8 and 9). We identified two (rs72646508, rs145777339) and six low frequency variants (AF < 0.01) from meta- and pooled analysis respectively from American and African ancestral groups (Table 1). Since the *All of Us* cohort is enriched for American (Hispanic) and African ancestral samples, we were able to identify multiple variants unique to these ancestral groups using the pooled approach. Among the ancestry-specific variants from the pooled analysis we identified 5 rare variants specific to African ancestry and 1 from American ancestry. We also observed that the 64 variants uniquely significant in pooled analysis had more significant CADD scores (Phred-scores >= 20) when compared to those uniquely significant in meta-analysis (p-value 0.02), with much of the signal observed in the American ancestral group (p-value 0.09). The variants identified from pooled analysis (Phred-scores >= 20) were rare and present in non-European ancestry and these variants harbored functional severe consequences extending to missense, frameshift and stop-gain mutations.

## Cost and complexity differences between pooled and meta-analyses

Cost and complexity are critical considerations impacting the use and usability of large-scale biomedical research data. We evaluated analysis complexity by examining the number of discrete computational steps required to complete a lipid GWAS (Fig. 1). The number of arrows (where each arrow represents an input or output of a computational step) required for the meta- and pooled analysis were 32 and 19, respectively. The increased complexity of the meta-analytical approach is primarily attributed to the duplication of computational steps within each silo. Extending this model to a theoretical analysis of N datasets siloed in N distinct TREs, the number of arrows required to complete the GWAS scales linearly at ~4x faster rate with the number of siloed TREs in the meta-analysis versus the pooled analysis (see supplement).

Additionally, we report the cost comparison of the meta- versus pooled analyses. There are two aspects to the overall cost: (1) Cloud resource utilization (including the cost of data storage and cloud compute), and (2) the person-time needed to perform and review the results of each step. For cloud data storage costs, the respective TREs assume the considerable cost of hosting the primary formats of the genomic data, freeing researchers of this cost burden. Cloud compute costs are tool dependent. For analysis steps involving R, PLINK, or

REGENIE the cloud compute resource costs are quite low - on the order of cents to a few dollars. Analysis steps involving Hail, by comparison, incur increased cloud compute cost. Hail processes data in a parallel fashion, leading to reduced wall-clock time to complete large-scale analyses. Hail is particularly useful whenever there does not already exist an optimized, purpose-built tool to perform the exact genomic data transformation needed. The primary cost driver for the meta-analysis was the Hail processing needed to extract relevant *All of Us* data from a Hail matrix table to create a BGEN file for use with REGENIE ($220). The primary cost driver for the pooled analysis was the Hail processing needed to merge the UK Biobank and *All of Us* variant data ($360).

Person-time is highly dependent on the researcher's familiarity with the datasets, methods, tools, and TRE capabilities. We found the amount of person-time for the meta-analyses was roughly twice that required for the pooled analyses. The person-time savings gained during pooled data harmonization, manipulation, and visualization within a single analysis environment, outweighed the cost of the additional steps required to merge the phenotype and genomic data.

## Discussion

We present two potential methods for the cross-analysis of UK Biobank and *All of Us* data using lipid GWAS as a case-study in computational approaches to analysis across TREs. Specifically, we looked at scientific and technical differences between meta-analysis of data in separate TRE silos, and pooled analysis of data in a single TRE. In each analysis we controlled for potential batch effects by including the source cohort as a covariate and limiting both pooled and meta-analyses to the subset of variants common in both the *All of Us* and UK Biobank cohorts. Each approach successfully replicated known genetic associations with plasma lipids. For both approaches, effect sizes found for each lipid trait are highly correlated with previously published studies. However, we did note several important scientific differences. First, pooled analysis enabled 1,496,404 additional variants to be included in the GWAS, compared with meta-analysis. Most of these variants were of lower minor allele frequencies, and thus this difference may be attributed to the fact that merging the two cohorts prior to applying the AC > 6 filter "rescued" rarer variants. We expect that the smaller overall number of variants retained for meta-analysis, because variants with an exceptionally low allele count are not considered by the analysis method, may negatively impact analysis of rare disease or rare variants. In these cases, a pooled approach may be preferred.

Second, the analytical approach impacted the number and ancestry frequency distributions of variants significantly associated with our phenotype of interest. We report 490 variants significantly associated with LDL-C from meta-analysis of GWAS performed separately in *All of Us* and UK Biobank TREs. In comparison, we found 464 variants significantly associated with LDL-C from pooled analysis of *All of Us* genome and UK Biobank exome sequencing data. We noted approximately 20% of variants significant in only the pooled analysis or significant in only the meta-analysis were most prevalent in non-European, non-Asian ancestry individuals. Prior foundational work has demonstrated that

**Table 2 | Important capabilities and opportunities to consider for improved cross-cohort analysis**

| | | |
|---|---|---|
| Data Access Safeguards | Existing Capability | - Maintain a single centrally funded copy of data that can be accessed in-place by researchers |
| | Opportunity | - Expand the ability to store temporary working data outside the source TRE (e.g., to create a single table containing all the multi-cohort phenotypes being studied)<br>- Engage with participants around the potential scientific value balanced by privacy and trust concerns of disseminating more granular results (e.g. results summarizing observations from <20 individuals without applying for an exception)<br>- Support mirroring of several datasets into one or more mutually trusted multi-dataset TREs<br>- Joint call the WGS data for the two cohorts, and make it available to researchers that have been granted access to both cohorts. |
| Research Support | Existing Capability | - Have a reasonable researcher-onboarding process and good researcher documentation on how to do in-TRE analysis |
| | Opportunity | - Build a library of cross-TRE-analysis examples, including run-it-yourself copies of well-documented analysis code, that cover a variety of analysis types and input datasets |
| Analysis Infrastructure | Existing Capability | - Support standard code packaging tools, especially Docker containers and Jupyter notebooks<br>- Provide flexible access to native cloud infrastructure, including different compute, storage, and database resources<br>- Provide access to large-scale analysis methods, including special-purpose tools like REGENIE and general-purpose tools like Hail |
| | Opportunity | - Provide access to a single dataset from more than one TRE and include mappings to common vocabularies or data models, to make it easier to share analysis code<br>- Use standard analysis application programming interfaces, such as those from the GA4GH, to allow central orchestration of distributed analysis using common methods<br>- Expose cloud-native data analysis tooling (vs. requiring researchers to learn and use TRE-specific tooling and techniques) |

given otherwise equivalent datasets pooled and meta-analysis will generate theoretically and empirically equivalent results[25,26]. However real-world experience as illustrated above and by others[27–29] has identified numerous differences between cohorts including phenotype ascertainment, genetic ancestry and population structure. Therefore, it is not surprising that these two analytical approaches yielded scientifically similar, but not identical, results. This has important implications for studying genetic variants in diverse individuals.

In addition to the scientific differences considered above, researchers seeking to analyze data across TREs face significant technical hurdles. Both complexity and cost scale with the number of data enclaves cross-analyzed. The pooled GWAS approach described was the least complex of the two investigated, requiring almost half as many discrete computational steps as meta-analysis. While analysis steps are displayed in a logical order in Fig. 1, many steps are run multiple times as an analyst becomes familiar with the datasets and capabilities of the respective TREs. The number of computational steps involved in meta-analysis grows at a ~4$x$ faster rate than for pooled, and therefore there is a significant increase in meta-analysis cost associated with the person-time required to develop and debug an analysis. That increased cost is high for two TREs, and even more significant as the number of TREs increases, which is expected as the amount of valuable global data increases.

This study found several capabilities provided by existing TREs that facilitated cross-cohort analysis, and that if adopted by future TREs would facilitate incorporation of more data into future analyses. These include: (1) maintaining a single centrally funded copy of data that can be accessed in-place by researchers, (2) providing robust, integrated research support, (3) providing access to flexible, scalable infrastructure and tools suited to large-scale data analysis (Table 2).

In addition, this study identified many opportunities to improve the support for cross-analysis in current and future TREs, including both technical and policy considerations (Table 2). In a meta-analysis, TRE technical differences (such as differences in user interfaces, analytical tools, supported programming languages, acceptable mechanisms for data access, acceptable mechanisms for data output, and methods for organizing and orchestrating an analysis) are considerable hurdles. The activation energy just to "get started" in multiple TREs is high. Our study team found it challenging to manage multiple copies of code in separate TREs. Data harmonization, a critical and time-consuming step, becomes much more tedious and error prone

when one cannot view and visualize together the row-level data. Many common analytical tasks, including creating a simple comparison plot with dots and whisker detail like the one in Fig. 3a, are infeasible with aggregate data. Improved harmonization and standardization of data, policies, and working environments across TREs can help reduce this burden.

Policy decisions are based on complex rationale that attempt to balance participant privacy, data security, scientific utility, and data sharing goals which have significant practical impact on cross-analysis. Policy changes that enable researchers to cross-analyze pooled data in one or more mutually trusted TREs would be a powerful step forward towards improved data usability and increased researcher productivity. The additional friction incurred when performing data harmonization for the meta-analysis could be reduced if TREs had reciprocal policies that permitted some participant level data, such as phenotypes, to be securely transferred between them. This middle-ground approach may be a compromise to increase data usability in a manner respectful of the current myriad of genomic data sharing policy and governance issues.

The analyses and results in this paper have several limitations. First, cross-analyses were limited to *All of Us* whole genome sequence and UK Biobank whole exome data available at the time of this study and meeting the TRE policy constraints. As noted previously, these data were generated using different sequencing methods and informatics pipelines. Future cross-analyses may be improved by further harmonizing approaches and joint-calling pipelines used to generate these data. The primary goal of this work was to build and describe approved paths for cross-analysis to encourage use by the broader scientific community. As such, the case study selected for cross-analysis was intentionally limited to common variants associated with well-studied lipid phenotypes. Future cross-analysis of *All of Us* and UK Biobank data exploring rare-variants and novel associations are likely to have greater scientific impact, and potentially to surface greater sensitivity to methodological differences. Finally, this study was limited to the cross-analysis of data residing in two enclaves. Future work is needed to expand these approaches to cross-analysis of data residing in three or more enclaves.

Early paths for cross-analysis of population-scale clinical and genomic data are clear. Program leaders, data providers, policy groups, and TRE developers have a shared responsibility to ensure data assets generated from public funding yield maximal scientific benefit while continuing to balance and honor participants as partners

in research programs. Thoughtful approaches to reducing barriers for efficient data access and analysis across large programs can increase the power of discovery while preserving participant trust. Data providers could consider providing mirrored copies of the data in multiple clouds to better enable pooled analyses. Additionally, and consistent with many existing efforts at federated analysis, data generators can further harmonize and standardize methods to avoid the need for downstream researchers to re-align and re-call genomic data. This study reinforces the need to reduce friction in cross-analysis to fully realize the potential of global-scale health research.

## Methods
### Cohorts
The UK Biobank (UKB) is a population-based cohort of approximately 500,000 participants recruited from 2006 to 2010, that has existing genomic and longitudinal phenotypic data. Baseline assessments were conducted at 22 assessment centers across the United Kingdom, with sample collections including blood-derived DNA. Secondary use of this data was approved by the Massachusetts General Hospital Institutional Review Board (protocol 2021P002228) and was facilitated through UK Biobank application 7089. The *All of Us* research program recruited individuals that have been and continue to be underrepresented in biomedical research due to limited access to healthcare. The first release of genomic data included approximately 98,000 individuals who completed electronic consent modules and health questionnaires upon enrollment. Approval to use the dataset for program operational demonstration projects was obtained from the *All of Us* Institutional Review Board.

### Genotypes
Whole exome sequencing (WES) from the 200 K exome release is the most recent release of genomic data permitted by UK Biobank policy to be analyzed outside of the UK Biobank Research Analysis Platform (RAP). The 200 K exome release includes approximately 10 Million exonic variants with >95% of targeted bases covered at a depth of 20X or greater. On both the *All of Us* Researcher Workbench (AoU RW) and the UK Biobank Research Analysis Platform (RAP), the genotypes were filtered to include only variants within the exome capture region with an alternative allele frequency of 6 or more. Whole genome sequenced (WGS) data from *All of Us* alpha 3 release was available as a Hail matrix table on the AoU RW. The alpha3 genotypes were filtered to include only variants within the same exome capture region with an alternative allele frequency of 6 or more. As initial quality control, variants with Hardy-Weinberg equilibrium exact test $p$-value below 1e-15 or missing call rates exceeding 10% were removed. QC also checked for samples with missing call rates exceeding 10%, but none were found. To mitigate batch effects, in the pooled analysis the prepared genotypes were filtered to include only those variants found in both cohorts and in the meta-analysis the results were filtered to include only those indicated found to be in both cohorts.

### Phenotypes
The primary outcomes in this study included LDL cholesterol (LDL-C), HDL cholesterol (HDL-C), total cholesterol (TC) and triglycerides (TG) as phenotypes. We curated and harmonized the lipid measurements and statin drug exposures for both UK Biobank and *All of Us* from the phenotype resources of these cohorts. LDL-C was either directly measured or calculated by the Friedewald equation when triglycerides were <400 mg/dL. Given the average effect of lipid lowering-medicines, when lipid-lowering medicines were present, we adjusted the total cholesterol by dividing by 0.8 and LDL-C by dividing by 0.7, triglycerides remained natural log transformed for analysis. The lipid phenotypes were then inverse rank normalized by the residuals, scaled by the standard deviation and adjusted for the covariates. We included PC1-10, age, age$^2$ and sex at birth as covariates in our study. To mitigate

batch effects, for the pooled analysis we also included a covariate of 'cohort'.

### Statistical analysis
Single variant genome wide association studies (GWAS) were carried out using REGENIE v2.2.4. We implemented REGENIE Step1 NULL model generation using quality-controlled variants with a minor allele count (MAC) of 100. We applied the leave one chromosome out (LOCO) method for GWAS while adjusting for the covariates stated above. We used variant and sample missingness at 10% followed by Hardy-Weinberg equilibrium $p$-value not exceeding $1 \times 10^{-15}$ for both step 1 and for the genome wide associations. We carried out meta-analysis of the siloed GWAS results from each cohort using the METAL package with the Standard Error scheme, where the methods weights effect size estimates using the inverse of the corresponding standard errors. The UKB siloed analysis was carried out on the UKB RAP, and the *All of Us* siloed analysis and the pooled analysis were carried out on the AoU RW. All the steps were implemented in R or Python notebooks. Complete details on the various steps carried out in the project are provided in the supplementary information.

### Reporting summary
Further information on research design is available in the Nature Portfolio Reporting Summary linked to this article.

## Data availability
The UK Biobank (UKB) whole-exome sequence data can be accessed through UKB Research Analysis Platform (RAP), through the UKB approval system (https://www.ukbiobank.ac.uk). Access to individual-level data from the *All of Us* research program is available to researchers whose institution has signed a data use agreement with *All of Us* (https://www.researchallofus.org/register/). Whole-genome sequencing data belongs to the controlled tier dataset, which requires additional training to access. gnomAD is publicly available (https://gnomad.broadinstitute.org/). The significant GWAS results generated in this study are provided in the Supplementary Data file.

## Code availability
The code for all analyses can be found in https://github.com/all-of-us/ukb-cross-analysis-demo-project[30] and was compatible with UK Biobank Research Analysis Platform and *All of Us* Researcher Workbench available data and technical capabilities as of the Spring of 2022.

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

## Acknowledgements

The *All of Us* Research Program is supported by the National Institutes of Health, Office of the Director: Regional Medical Centers: 1 OT2 OD026549; 1 OT2 OD026554; 1 OT2 OD026557; 1 OT2 OD026556; 1 OT2 OD026550; 1 OT2 OD 026552; 1 OT2 OD026553; 1 OT2 OD026548; 1 OT2 OD026551; 1 OT2 OD026555; IAA #: AOD 16037; Federally Qualified Health Centers: HHSN 263201600085U; Data and Research Center: 5 U2C OD023196; Genome Centers: OT2OD002748, OT2OD002750, OT2OD002751; Biobank: 1 U24 OD023121; The Participant Center: U24 OD023176; Participant Technology Systems Center: 1 U24 OD023163; Communications and Engagement: 3 OT2 OD023205; 3 OT2 OD023206; and Community Partners: 1 OT2 OD025277; 3 OT2 OD025315; 1 OT2 OD025337; 1 OT2 OD025276. In addition, the *All of Us* Research Program would not be possible without the partnership of its participants. The authors greatly appreciate feedback from Dr. Paul Harris on an early draft of the manuscript. P.N. is supported by grants from NHLBI/NIH (R01HL142711, R01HL127564) and NHGRI/NIH (U01HG011719). AGB is supported by grants from NIH (DP5 OD029586) and a Burroughs Wellcome Fund Career Award for Medical Scientists. This research has been conducted using data from UK Biobank, a major biomedical database, under application number 7089. *All of Us*, the *All of Us* logo, and "The Future of Health Begins with You" are service marks of the U.S. Department of Health and Human Services. This content is solely the responsibility of the authors and does not necessarily represent the official views of the National Institutes of Health.

## Author contributions

N.D., M.S.S., K.M., C.L., A.A.P., D.M.R., J.C.D., M.E., D.G., P.N. and A.G.B. contributed to the conception and design of the work. N.D. and M.S.S. performed the formal analysis. N.D., M.S.S., H.R.C., K.M., and S.H. contributed to the data acquisition and curation. K.M., M.A.B., C.L., A.M., R.C., N.A., M.E. and D.G. provided guidance on the datasets used in the study. N.D., M.S.S., K.M. and A.G.B. drafted the manuscript which was critically revised with contributions and input from all authors.

## Competing interests

P.N. reports investigator-initiated grants from Amgen, Apple, AstraZeneca, Boston Scientific, and Novartis, personal fees from Apple, Astra-Zeneca, Blackstone Life Sciences, Foresite Labs, Novartis, Roche/Genentech, is a co-founder of TenSixteen Bio, is a shareholder of geneXwell and TenSixteen Bio, and spousal employment at Vertex, all unrelated to the present work. A.G.B. is a co-founder and shareholder of TenSixteen Bio unrelated to the present work. N.D. and D.G. are employees of Verily Life Sciences and may own stock as part of the standard compensation package. A.P. serves as a Google Ventures (GV) venture partner and holds an equity interest in certain of GV's affiliated investment funds. A.P. has also received funding from Verily, MSFT, Intel, IBM, Bayer, Pfizer, Astra Zeneca, and Biogen. The remaining authors declare no competing interests.
