## [Peer Review File · Nature Communications]

Demonstrating paths for unlocking the value of cloud genomics through cross cohort analysisREVIEWER COMMENTS

Reviewer #1 (Remarks to the Author):

This is a technical paper focused on a surprisingly important technical task which is whether the meta-analysis of results from cohorts on separate TREs give similar or near identical results compared to pooled analysis. Broadly there is an important conclusion, which is this is not the case in the details, and the details matter particularly for individuals which are not well represented in the cohorts, often more diverse individuals in the cohorts.

However there is to my mind a major flaw, which is that a key restriction in the analysis was the policy based decision to have an allele count restriction of 40. It is unclear to me whether the reading of the All of Us policy for Pvalues needs to adhere to this (clearly this is wise for aggregate data) but importantly I believe the authors should explore the impact of doing meta-analysis at the limit of individual analysis (eg, the allele count of 6) and if that combination is closer compared to the policy enforced allele count of 40.

I cannot recommend publication without this being explored. Fundamentally the question is whether the barriers are science lead or policy lead in the meta-analysis deficit; if they are policy lead there might be other ways to fix this (eg, thinking aloud, special permission or encryption for sharing specific pvalue files which are then deleted afterwards by the meta-analysis team).

The second somewhat ugly aspect is that one of the analyses did not converge. Usually it does not converge due to low case numbers, but as this is continuous I am surprised. I do think the authors can just give up here - either one needs to delete this analysis (it is not core to my reading of the paper) or to find a work around; as frustrating as this is, this is the frustration of in practice GWAS (whether pooled or meta-analysis).

There other minor aspects of the paper:

There is no methods section in the main paper that I can see, and an extensive supplement. This is fair, but I think readers would like an abbreviated methods section in the main paper with the key choices

I would have in the flow diagram the total number of variants in pooled analysis and then the restriction via allele count 6 as separate sections

Figure 3 has poor resolution and truncated axes (at least for me!). Surely this can be far better.

Reviewer #2 (Remarks to the Author):

This manuscript discuss how to utilized the cloud-based trusted research environment (TRE) for individual genotype and phenotype data analysis including GWAS, WS, and WGS. The WGS data from All of Us (n=98,000) and WES data from UK Biobank (n=200,000) were utilized. Authors conducted a comparative survey of the pooled analysis where the two datasets are in the same place and the meta-analysis where the datasets are at the different places. The GWAS analysis for the LDL level were conducted in parallel for both methods, which yielded the similar numbers of the associated variants. Authors then conducted a series of comparisons including the associated variant annotation and cost for analysis. This manuscript focuses on an important topic in the biobank-based genome research. This reviewer has several comments.

1. The distributions of the causal variants in the allele frequency spectra, population specificity, and functional annotations depend on the phenotype of interests. Authors should analysis more numbers of the phenotypes rather than only LDL.
2. In the GWAS, authors used REGENIE, but other methods should also be applied such as SAIGE. We know that REGENIE outputs slightly different association results even when adopting the same regression model.
3. One important merit of the pooled analysis is joint calling of the variants. Joint calling integrating the larger samples can provide more accurate estimates of the variants, especially in non-SNV variants such as CNV or SV. This must be assessed.
4. The analysis only focused on the variants which were shared between the datasets. Thus, the conclusion from this analysis is currently only applicable to the WES, but not to GWAS or WGS. This limitations should be assessed.
5. As for the discussions on the GWAS results comparisons, the lead variants after fine-mapping are recommended rather than the variants that simple satisfied the significance threshold.
6. Discussions on the increased complexity, person-time, and cost in the meta-analysis is interesting. These burdens become apparent when a larger number of the TREs are utilized separately for the datasets more than two. Discussions on this point is recommended.
7. Since this study focuses on the WES analysis, gene-based rare variant test (both one-sided and two-sided tests) should be assessed. Meta-analysis of the gene-based test results is one of the challenging issues, so should be a best topic for the pooled vs meta comparative analysis in the manuscript.
8. Beyond cost, fairness and transparency in data access management is the most important point. Centralized TRE can have the risk of data monopolization without strong policy for open science.

REVIEWER COMMENTS

Reviewer #1 (Remarks to the Author):

This is a technical paper focused on a surprisingly important technical task which is whether the meta-analysis of results from cohorts on separate TREs give similar or near identical results compared to pooled analysis. Broadly there is an important conclusion, which is this is not the case in the details, and the details matter particularly for individuals which are not well represented in the cohorts, often more diverse individuals in the cohorts.

However there is to my mind a major flaw, which is that a key restriction in the analysis was the policy based decision to have an allele count restriction of 40. It is unclear to me whether the reading of the All of Us policy for Pvalues needs to adhere to this (clearly this is wise for aggregate data) but importantly I believe the authors should explore the impact of doing meta-analysis at the limit of individual analysis (eg, the allele count of 6) and if that combination is closer compared to the policy enforced allele count of 40.

I cannot recommend publication without this being explored. Fundamentally the question is whether the barriers are science lead or policy lead in the meta-analysis deficit; if they are policy lead there might be other ways to fix this (eg, thinking aloud, special permission or encryption for sharing specific pvalue files which are then deleted afterwards by the meta-analysis team).

Thank you for this positive feedback. We submitted an exception request to the *All of Us* Research Program Resource Access Board (RAB), which was approved, and we can now download GWAS results without filtering for use in the meta-analysis and disseminate results below AC \geq 40.

The manuscript has been updated to reflect meta-analysis results for AC \geq 6. We also found and fixed an issue with variant ids for indels that had caused them to appear as single cohort results in the meta-analysis. Overall the central findings of this work remain the same whether using AC \geq 6 or AC \geq 40.

Flowchart for AC \geq 6

Previous flowchart for AC >= 40, for comparison

The second somewhat ugly aspect is that one of the analyses did not converge. Usually it does not converge due to low case numbers, but as this is continuous I am surprised. I do think the authors can just give up here - either one needs to delete this analysis (it is not core to my reading of the paper) or to find a work around; as frustrating as this is, this is the frustration of in practice GWAS (whether pooled or meta-analysis).

Thank you for this feedback. We have removed those results per your suggestion.

There other minor aspects of the paper:

There is no methods section in the main paper that I can see, and an extensive supplement. This is fair, but I think readers would like an abbreviated methods section in the main paper with the key choices

We have now added a brief methods section to the main paper.

I would have in the flow diagram the total number of variants in pooled analysis and then the restriction via allele count 6 as separate sections

Thank you for this feedback. We have updated Figure 2.

Figure 3 has poor resolution and truncated axes (at least for me!). Surely this can be far better.

Thank you for this feedback. We have corrected the resolution in Figure 3.

a LDL per person by cohort, before and after adjustment for statin use

Reviewer #2 (Remarks to the Author):

This manuscript discuss how to utilized the cloud-based trusted research environment (TRE) for individual genotype and phenotype data analysis including GWAS, WS, and WGS. The WGS data from All of Us (n=98,000) and WES data from UK Biobank (n=200,000) were utilized. Authors conducted a comparative survey of the pooled analysis where the two datasets are in the same place and the meta-analysis where the datasets are at the different places. The GWAS analysis for the LDL level were conducted in parallel for both methods, which yielded the similar numbers of the associated variants. Authors then conducted a series of comparisons including the associated variant annotation and cost for analysis. This manuscript focuses on an important topic in the biobank-based genome research. This reviewer has several comments.

1. The distributions of the causal variants in the allele frequency spectra, population specificity, and functional annotations depend on the phenotype of interests. Authors should analysis more numbers of the phenotypes rather than only LDL.

Thank you for this feedback. We have analyzed other lipids including HDL-C, TC and TG along with LDL-C. We have documented the summary statistics allele frequency and population specificity to the lead variants. Please see the summary results in Supplementary Tables 2 and 3 for meta-analysis and pooled analysis, respectively. Overall, the findings across these three other traits are highly concordant with the analyses of LDL-C.

2. In the GWAS, authors used REGENIE, but other methods should also be applied such as SAIGE. We know that REGENIE outputs slightly different association results even when adopting the same regression model.

Thank you for this feedback. Please see the comparison of pooled SAIGE and REGENIE results for LDL-C within chromosome 19. The association results are slightly different, as you note in your feedback, however with a R-square value of 0.88 we believe the results we present here are not dependent on the particular genetic association tool.

3. One important merit of the pooled analysis is joint calling of the variants. Joint calling integrating the larger samples can provide more accurate estimates of the variants, especially in non-SNV variants such as CNV or SV. This must be assessed.

Under the current policies of UK Biobank and All of Us, researchers are unable to pool WGS data, and therefore cannot perform joint calling themselves. It would be of great value to the research community for the UK Biobank and All of Us TREs to agree to jointly-call their cohorts, and provide that jointly-called data to researchers who have access to both cohorts. We have added this TRE opportunity to Table 2 in the revised manuscript.

Text added in the revised manuscript:

- Joint call the WGS data for the two cohorts, and make it available to researchers that have been granted access to both cohorts.

4. The analysis only focused on the variants which were shared between the datasets. Thus, the conclusion from this analysis is currently only applicable to the WES, but not to GWAS or WGS. This limitations should be assessed.

Thank you for this feedback. Under the current policies of UK Biobank and All of Us, researchers are unable to pool the WGS data, therefore we are unable to assess this particular limitation because of external constraints on the two resources.

5. As for the discussions on the GWAS results comparisons, the lead variants after fine-mapping are recommended rather than the variants that simply satisfied the significance threshold.

Thank you for this feedback. We have identified the lead variants by clumping the significant variants from each lipid (Supplemental Tables 2 and 3). We carried out gene prioritization of the variants from our analysis for LDL-C phenotype using POPS methodology and observed all important genes to be fine mapped using our results. We have included a sentence in the Results section of the revised manuscript.

Text added in the revised manuscript:

Gene prioritization of the GWAS results from our analysis, fine-mapped variants to genes important to lipids including *APOE*, *APOA2*, *LDLR*, *PCSK9*, *CEPT*, *APOA5*, *APOB* with top 20 prioritization scores.

6. Discussions on the increased complexity, person-time, and cost in the meta-analysis is interesting. These burdens become apparent when a larger number of the TREs are utilized separately for the datasets more than two. Discussions on this point is recommended.

Thank you for this feedback. Please see the additional mention of the ~4x growth rate added to the Discussion and the elaboration on the increased amount of person-time.

Text added in the revised manuscript:

The number of computational steps involved in meta-analysis grows at a ~4x faster rate than for pooled, and therefore there is a significant increase in meta-analysis cost associated with the person-time required to develop and debug an analysis. That increased cost is high for two TREs, and even more significant as the number of TREs increases, which is expected as the amount of valuable global data increases.

Future work is needed to expand these approaches to cross-analysis of data residing in **three or more** enclaves.

7. Since this study focuses on the WES analysis, gene-based rare variant test (both one-sided and two-sided tests) should be assessed. Meta-analysis of the gene-based test results is one of the challenging issues, so should be a best topic for the pooled vs meta comparative analysis in the manuscript.

Thank you for this feedback. We very much agree with the need for assessment with respect to rare variant analysis. We are working on a companion manuscript to explore this in greater detail. Additionally, a recent paper using *All of Us* WGS data with rare variants has been published (referred below) and it shows significant rare-variant burden estimates with LDL-C phenotype.

Common and rare variants associated with cardiometabolic traits across 98,622 whole-genome sequences in the *All of Us* research program. Xin Wang *et al.* *Journal of Human Genetics* (2023)

8. Beyond cost, fairness and transparency in data access management is the most important point. Centralized TRE can have the risk of data monopolization without strong policy for open science.

Thank you for this feedback. Please see the updates to Table 2 and the Discussion indicating opportunity for multiple mutually trusted multi-dataset TREs.

Text added in the revised manuscript:

- Support mirroring of several datasets into **one or more** mutually trusted multi-dataset **TREs**

Policy changes that enable researchers to cross-analyze pooled data in **one or more** mutually trusted **TREs** would be a powerful step forward towards improved data usability and increased researcher productivity.

REVIEWERS' COMMENTS

Reviewer #1 (Remarks to the Author):

Nice revision. Interesting that the issues remained at AC >6

Reviewer #2 (Remarks to the Author):

Authors fully responded the comments.